energy/materials science

dopamine, MWCNT, nitrogen doping, vanadium redox flow battery

**Author for correspondence:**
Hong Sun
e-mail: sunhongwxh@sina.com

This article has been edited by the Royal Society of Chemistry, including the commissioning, peer review process and editorial aspects up to the point of acceptance.

# Dopamine-derived nitrogen-doped carboxyl multiwalled carbon nanotube-modified graphite felt with improved electrochemical activity for vanadium redox flow batteries

Qiang Li, Anyu Bai, Tianyu Zhang, Song Li and Hong Sun

School of Mechanical Engineering, Shenyang Jianzhu University, Shenyang 110168, People's Republic of China

 QL, 0000-0002-2672-0917; HS, 0000-0001-6799-6015

Improving the electrochemical activity of electrodes is essential to the development of vanadium redox flow battery (VRFB). In this work, we prepared a novel electrode with the modification of nitrogen-doped carboxyl multiwalled carbon nanotubes using dopamine as an eco-friendly nitrogen source (carboxyl MWCNT@PDAt). Characterization and electrochemical measurements reveal that the synthesized carboxyl MWCNT@PDAt-modified graphite felt electrode (carboxyl MWCNT@PDAt/GF) exhibits excellent electrochemical performance toward $VO^{2+}/VO_2^+$ reaction. Superior battery performance was obtained with the energy efficiency of 80.54% at a current density of 80 mA cm$^{-2}$. Excellent durability of the carboxyl MWCNT@PDAt/GF electrode was confirmed by long-term charge/discharge tests. The enhanced reaction kinetics of $VO^{2+}/VO_2^+$ is ascribed to the synergetic effect of oxygen and nitrogen containing groups on graphite felt surface and the presence of nitrogen-doped carboxyl multiwalled carbon nanotubes (MWCNT). The facile approach proposed in this paper provides a new route to the fabrication of electrode with excellent performance for VRFB.

# 1. Introduction

Renewable energies, including hydroenergy, geothermal energy and solar energy, have drawn great attention owing to the growing energy demand and environmental hazards. However, due to their inherent random and intermittent nature, renewable energies require integration with energy storage system to improve their utilization. Hence, great efforts have been devoted to the development of energy storage system. As one of the promising energy storage systems, vanadium redox flow battery (VRFB) has drawn great attention due to its unique features of long life cycles, design flexibility and safety [1,2]. VRFB employs $VO^{2+}/VO_2^+$ and $V^{3+}/V^{2+}$ as positive and negative redox reaction, respectively. Thus, the cross contamination caused by different metal ions can be effectively eliminated [3]. Carbon-based electrodes, taking the advantages of long cycle life under acidic solution, high conductivity and low cost for commercial application, have been widely investigated in the VRFB area over the past few decades [4]. Currently, the most widely used carbon-based electrodes in VRFB are graphite or carbon felts, due to their porous structures, high conductivities and corrosion resistance in highly acidic electrolytes [5]. However, the graphite or carbon felts suffer from the problems of low electrochemical activity, which impedes the enhancement of the energy efficiency and capacity of VRFB, making it difficult to meet the demands of energy storage system [6]. Furthermore, it is worth pointing out that the limitation of VRFB performance is primarily ascribed to the relatively lower kinetics of $VO^{2+}/VO_2^+$ reaction, while the kinetics of $V^{2+}/V^{3+}$ reaction in the negative part is faster [7]. In order to overcome the poor electrochemical activity toward $VO^{2+}/VO_2^+$ reaction, great efforts have been devoted to the development of novel electrode by modifying electrodes and/or introducing catalysts [8–12].

Among various treatments, oxygen or nitrogen containing functional groups have been confirmed to efficiently enhance the reaction kinetics of VRFB [13–17]. It has been demonstrated that the introduction of oxygen or nitrogen containing functional groups provides more active sites for vanadium ions reaction and facilitates the adsorption of vanadium ions, resulting in an improvement in the catalytic activity of electrode [18–20]. Recently, carbon nanotubes (CNTs) have drawn a great deal of attention in VRFB area due to their high electrical conductivity, superior chemical stability in acid solution and high specific surface area [3,21,22]. Nitrogen-doped CNTs were used as catalysts and exhibited enhanced performance for VRFB [23–25]. The composite electrode with the modification of N-doped CNTs not only shows large specific surface area, but also shows excellent electrochemical activity and reversibility [26]. Due to the introduced N atoms in carbon framework, more free electrons could be ionized out and more defects could be produced, resulting in a significant promotion in the vanadium ions transport during charge/discharge operation [27–29].

However, the proposed methods for the preparation of nitrogen-doped CNTs are complicated. Furthermore, toxic reagents are usually required, which limits the wide application of nitrogen-doped CNTs. Dopamine, which is non-toxic, has shown versatile coating capabilities on surface of various materials, yielding high concentration of amine groups [30,31]. Dopamine has been used as a nitrogen source for the fabrication of nitrogen-doped electrode in VRFB. The dopamine-derived composite electrode shows significant improvement in cell performance [32,33]. Especially, there have been limited reports on the synergistic effect between nitrogen doping and oxygen containing groups modification. Hence, a facile and practical process for the fabrication of graphite felt (GF) with nitrogen doping and oxygen containing groups modification remains a priority in the development of VRFB.

In this study, the effect of carboxyl MWCNT@PDAt on the performance of VRFB was systematically analysed. GF electrode with the modification of carboxyl MWCNT@PDAt was prepared and characterized using various analytical techniques including scanning electron microscope (SEM), four-point probe test, Brunauer–Emmett–Teller (BET) surface area measurement, Raman spectroscopy (Raman) and X-ray photoelectron spectroscopy (XPS). The effect of carboxyl MWCNT@PDAt on the reaction kinetics of $VO^{2+}/VO_2^+$ couple was investigated based on cyclic voltammetry (CV) and electrochemical impedance spectroscopy (EIS). Charge/discharge measurements were also conducted to evaluate the efficiency and stability of carboxyl MWCNT@PDAt/GF in acid solution.

# 2. Experimental section

## 2.1. Material and methods

The chemicals, including dopamine hydrochloride (98%, Sigma-Aldrich), carboxyl multiwalled carbon nanotubes (MWCNT) (greater than 95%, Aladdin), sulfuric acid, hydrated vanadium sulfate and ethanol, were used in this paper. All chemicals were analytical reagent grade and directly used for the preparation of composite electrode. Polyacrylonitrile (PAN) GFs, purchased from Sigracell SGL Inc.,

were used as electrodes. Nafion 117 purchased from Du-Pont Inc. was used as a proton exchange membrane in an assembled home-made single cell.

Prior to use, pieces of PAN GFs were washed with deionized water and ethanol three times. The pre-processed GFs were dried in an oven at 80°C for 6 h. The polydopamine-modified carboxyl MWCNTs without heat treatment (carboxyl MWCNT@PDA) were prepared based on literature [34]. The detailed synthetic process of carboxyl MWCNT@PDA is given in the electronic supplementary material. Then, 50 mg prepared carboxyl MWCNT@PDA were added into a 50 ml N, N-dimethylformamide (DMF) solution under ultrasonic process for 1 h. Then, the suspension was further processed by magnetic stirring to form a uniformly dispersed solution. The GFs were immersed in the suspension at room temperature for 24 h. Then, the prepared carboxyl MWCNT@PDA-modified GFs were washed with deionized water and ethanol three times, followed by drying in an oven at 60°C for 12 h. Finally, the prepared composite electrodes were treated under nitrogen atmosphere in a tube furnace. The as-prepared carboxyl MWCNT@PDA-modified GFs were heat treated at 500, 700 and 900°C for 5 h. The obtained composite GF electrodes under different heat treatment temperature were named as carboxyl MWCNT@PDAt-500/GF, carboxyl MWCNT@PDAt-700/GF and carboxyl MWCNT@PDAt-900/GF, respectively. For comparison, pristine carboxyl MWCNT without the adhesion of PDA were decorated on GF surface, followed by heat treatment at 900°C for 5 h. The resulting sample was denoted as MWCNT-900/GF and used for comparison with the performance of the carboxyl MWCNT@PDAt-modified GF electrode.

## 2.2. Material and methods

The surface morphology of prepared electrode was observed by SEM (Hitachi, S4800). The electronic conductivity of composite electrode was evaluated via four-probe method (RTS-8, 4 Probes Tech). The surface area was measured by BET method based on $N_2$ adsorption–desorption isotherm (TriStar II 3020, Micromeritics). The contents of nitrogen and oxygen elements in the modified electrode were characterized by XPS. The XPS characterization was conducted by an EscaLab 250XI system using Al-Kα radiation at a power of 350 W. The microstructures of the GF before and after the modification of carboxyl MWCNT@PDAt were tested by Raman spectra (XploRA™ PLUS, HORIBA).

## 2.3. Electrochemical tests

A typical three-electrode electrochemical cell, including a saturated calomel electrode as a reference electrode, a Pt electrode as a counter electrode and a piece of prepared carboxyl MWCNT@PDAt-modified electrode (1 × 1 cm) as a working electrode, was established for CV and EIS tests. All tests were conducted on an electrochemical workstation (Parstat 4000A, AMETEK) in a positive electrolyte of 0.1 M $VO^{2+}$ in 2 M $H_2SO_4$. CV plots were obtained at a scan rate of 5 mV s$^{-1}$ in potential range of 0–1.4 V. EIS measurements were performed with frequency range of 0.01 to $1 \times 10^5$ Hz at an AC perturbation voltage of 10 mV.

## 2.4. Vanadium redox flow battery performance

The influence of the prepared electrode on VRFB performance was evaluated by charge/discharge tests. The prepared carboxyl MWCNT@PDAt-modified electrode (3 × 3 cm) was used as positive electrode during VRFB operation. The electrolytes for catholyte and anolyte were 20 ml of 1.5 M $VO^{2+}$ + 4.2 M $H_2SO_4$ and 20 ml of 1.5 M $V^{3+}$ + 4.2 M $H_2SO_4$, respectively. The electrolytes were circulated through the VRFB cell by a peristaltic pump with a flow rate of 30 ml min$^{-1}$. All charge/discharge tests were conducted under galvanostatic condition over a voltage window of 0.8–1.65 V.

# 3. Results and discussion

## 3.1. Scanning electron microscope test

The surface morphology of carboxyl MWCNT@PDAt-900/GF was characterized by SEM. As shown in figure 1a, pristine GF shows smooth surface before treatment. But the surface of GF become rough due to the modification of carboxyl MWCNT@PDAt-900. The increase in the roughness of GF results in a larger surface area, which provides more active sites for electrochemical reaction. The results also

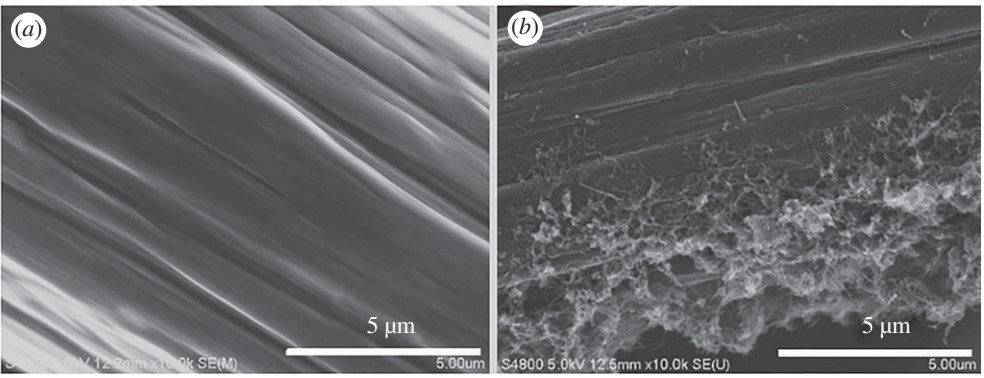

**Figure 1.** Surface morphology of composite electrode before (*a*) and after (*b*) the modification of carboxyl MWCNT@PDAt-900.

**Table 1.** Conductivities of pristine GF and types of carboxyl MWCNT@PDAt-modified electrodes.

|  | pristine GF | carboxyl MWCNT@PDAt-500/GF | carboxyl MWCNT@PDAt-700/GF | carboxyl MWCNT@PDAt-900/GF |
|---|---|---|---|---|
| conductivity (S cm$^{-1}$) | 19.30 | 21.60 | 22.92 | 25.79 |

**Table 2.** The BET surface area of pristine GF and carboxyl MWCNT@PDAt-modified electrodes.

|  | pristine GF | carboxyl MWCNT@PDAt-500/GF | carboxyl MWCNT@PDAt-700/GF | carboxyl MWCNT@PDAt-900/GF |
|---|---|---|---|---|
| specific surface area (m$^2$ g$^{-1}$) | 0.92 | 2.77 | 2.83 | 3.40 |

demonstrate that the carboxyl MWCNT@PDAt-900 have been successfully decorated on the GF electrode after heat treatment.

## 3.2. Conductivity measurement

To evaluate the variation of conductivity after modification, the conductivities of composite electrodes were tested by four-probe tester. The conductivities of pristine GF and composite electrodes are presented in table 1. It is obvious that the conductivities of composite electrodes are higher than that of pristine GF. Additionally, the conductivity increases with an increase in thermal treatment temperature. The carboxyl MWCNT@PDAt-900/GF shows the highest conductivity of 25.79 S cm$^{-1}$. The increase in conductivity of the composite electrode indicates that the electron transfer resistance decreases after the modification of carboxyl MWCNT@PDAt, which is a benefit for the decrease in Ohmic polarization during charge/discharge process.

## 3.3. Specific surface area test

The test of specific surface area was carried out by Micromeritics TriStar II 3020 based on BET method. The specific surface areas of carboxyl MWCNT@PDAt-modified electrodes are listed in table 2. In comparison with pristine GF, all MWCNT@PDAt-modified electrodes show increased surface area. The specific surface areas of pristine GF, carboxyl MWCNT@PDAt-500/GF and carboxyl MWCNT@PDAt-700/GF are 0.92, 2.77 and 2.83 m$^2$ g$^{-1}$, respectively. Compared with other samples, the carboxyl MWCNT@PDAt-900/GF shows the highest specific surface area of 3.40 m$^2$ g$^{-1}$, demonstrating that the specific surface area has been significantly increased after the decoration of carboxyl MWCNT@PDAt treated at 900°C.

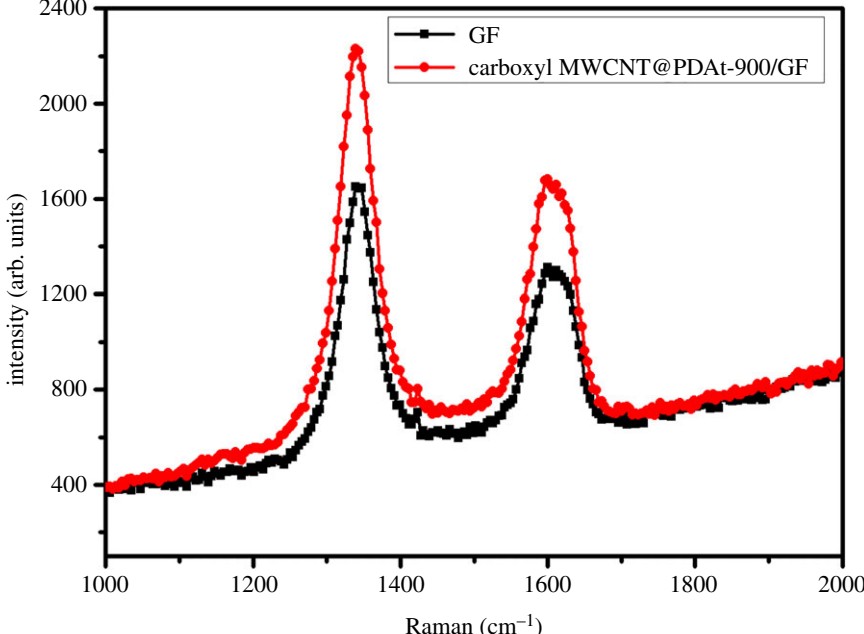

**Figure 2.** Raman spectra of pristine GF and carboxyl MWCNT@PDAt-900/GF.

## 3.4. Raman test

The microstructure of the prepared composite electrode was assessed by Raman spectra. The Raman spectra of GFs electrodes before and after the modification of carboxyl MWCNT@PDAt are shown in figure 2. Obvious D peak and G peak are observed in the Raman spectra, which are located at 1340 and 1600 cm$^{-1}$, respectively. The D peak is associated with the vibrational stretching of sp$^3$ defect site, while the G peak is related with the vibrational stretching of sp$^2$ carbons. The ratio of the intensity of D peak to that of G peak ($I_D/I_G$) represents the disorder degree of carbon-based materials. The higher value of $I_D/I_G$ indicates the higher disorder degree of carbon-based materials. According to figure 2, the calculated values of $I_D/I_G$ for pristine GF and carboxyl MWCNT@PDAt-900/GF are 1.27 and 1.35, respectively. Hence, the Raman spectra indicate that the disorder degree of electrode is increased after the modification of carboxyl MWCNT@PDAt-900 on GF surface, leading to an increase in the introduction of defects which are more active for the attraction of vanadium ions.

## 3.5. X-ray photoelectron spectroscopy test

The XPS spectra of carboxyl MWCNT, polydopamine-modified carboxyl MWCNT before heat treatment (carboxyl MWCNT@PDA) and carboxyl MWCNT@PDAt-900 are exhibited in figure 3. The elemental content of C, O and N are listed in table 3. From figure 3 and table 3, it is seen that the carboxyl MWCNT@PDA has more oxygen- and nitrogen-functional groups than carboxyl MWCNT, showing the oxygen content of 14.03% and nitrogen content of 8.98%. However, the content of oxygen and nitrogen for MWCNT@PDAt-900 decrease to 2.44% and 3.30%, respectively. Compared with carboxyl MWCNT@PDA, the carboxyl MWCNT@PDAt-900 shows the lower content of oxygen and nitrogen elements, indicating that the oxygen and nitrogen groups decompose during heat treatment process. The content of nitrogen for carboxyl MWCNT@PDAt-900 significantly increases in comparison with that of pristine carboxyl MWCNT, indicating that nitrogen atoms have been successful doped into carbon framework using dopamine as a nitrogen source.

Figure 4 displays the O1s spectra of carboxyl MWCNT, carboxyl MWCNT@PDA and carboxyl MWCNT@PDAt-900. All O1s spectra can be divided into two peaks. The binding energy at 531.1 eV corresponds to C=O bond, and the binding energy at 532.7 eV relates to C–O bond, respectively [35,36]. The fraction of C=O and C–O bonds are listed in table 4. The results in table 4 demonstrate that the content of C=O and C–O bonds in carboxyl MWCNT are 1.02% and 1.69%, while that in carboxyl MWCNT@PDAt-900 are 0.78% and 1.66%, respectively. No significant difference is found in the content of oxygen between carboxyl MWCNT and carboxyl MWCNT@PDAt-900. The content of C=O and C–O

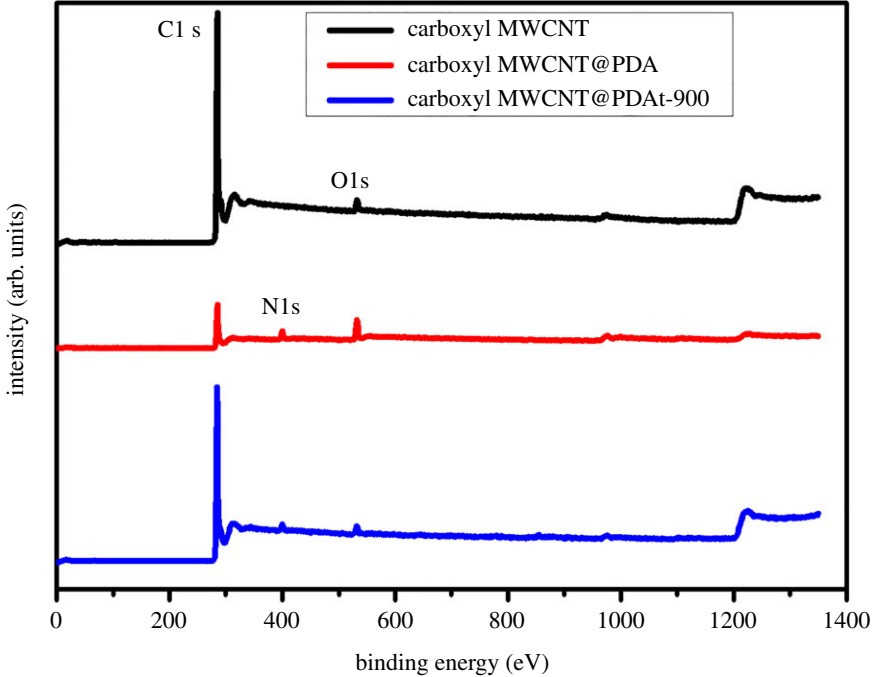

**Figure 3.** XPS spectra of carboxyl MWCNT, carboxyl MWCNT@PDA and carboxyl MWCNT@PDAt-900.

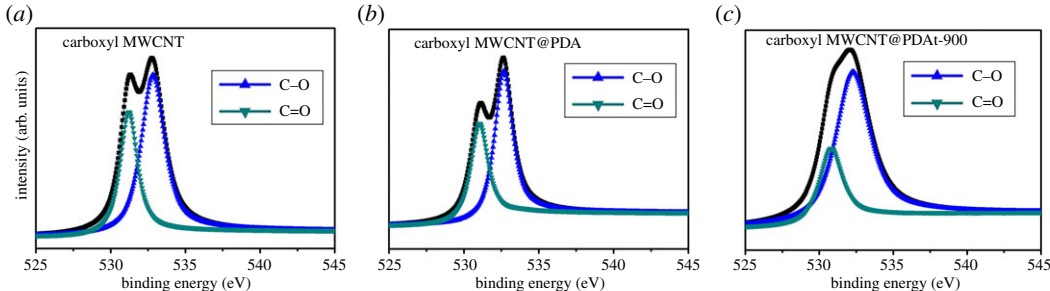

**Figure 4.** Curve-fit of O1s spectra from carboxyl MWCNT, carboxyl MWCNT@PDA and carboxyl MWCNT@PDAt-900.

**Table 3.** C, O and N atom content of carboxyl MWCNT, carboxyl MWCNT@PDA and carboxyl MWCNT@PDAt-900.

| samples | C (%) | N (%) | O (%) |
|---|---|---|---|
| carboxyl MWCNT | 96.87 | 0.41 | 2.72 |
| carboxyl MWCNT@PDA | 76.99 | 8.98 | 14.03 |
| carboxyl MWCNT@PDAt-900 | 94.26 | 3.30 | 2.44 |

**Table 4.** O1s and N1s XPS spectral fitting peak content of carboxyl MWCNT, carboxyl MWCNT@PDA and carboxyl MWCNT@PDAt-900.

| samples | C–O (%) | C=O (%) | pyridinic-N (%) | pyrrolic-N (%) | graphitic-N (%) |
|---|---|---|---|---|---|
| carboxyl MWCNT | 1.69 | 1.02 | — | — | — |
| carboxyl MWCNT@PDA | 8.40 | 5.63 | 2.60 | 4.86 | 1.52 |
| carboxyl MWCNT@PDAt-900 | 1.66 | 0.78 | 0.53 | 1.81 | 0.96 |

bonds in carboxyl MWCNT@PDA are 5.63% and 8.40%, respectively, indicating that the polydopamine (PDA) has been successfully absorbed on the surface of carboxyl MWCNT. Compared with carboxyl MWCNT@PDA, the content of C=O and C–O bonds in carboxyl MWCNT@PDAt-900 decrease, demonstrating the decomposition of oxygen containing groups under high temperature environment.

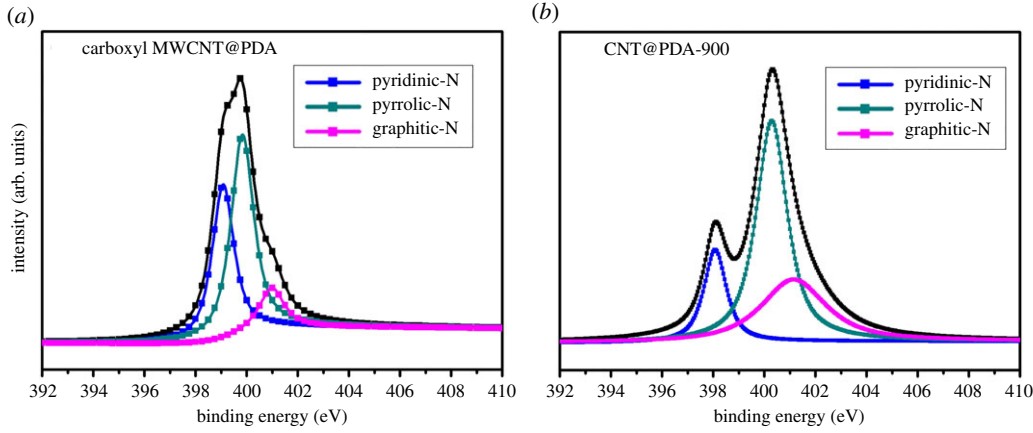

**Figure 5.** Curve-fit of N1s spectra from carboxyl MWCNT@PDA and carboxyl MWCNT@PDAt-900.

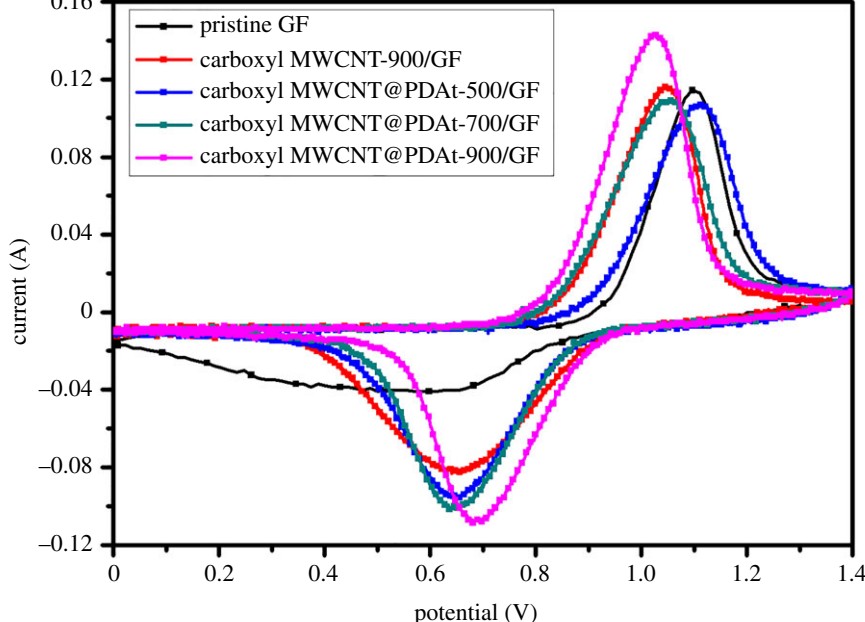

**Figure 6.** CV curves of pristine GF, carboxyl MWCNT-900/GF and carboxyl MWCNT@PDAt-modified electrodes with different treatment temperatures.

As seen in figure 5, the N1 s core-level spectra of carboxyl MWCNT@PDA and carboxyl MWCNT@PDAt-900 were deconvoluted into three different peaks, corresponding to pyridinic-N at 398.2 eV, pyrrolic-N at 400.1 eV and graphitic-N at 401.3 eV [18]. As shown in table 4, the content of pyridinic-N, pyrrolic-N and graphitic-N in carboxyl MWCNT@PDA is 2.60%, 4.86% and 1.52%, respectively. While for carboxyl MWCNT@PDAt-900, the corresponding content is 0.53%, 1.81% and 0.96%, respectively. The results demonstrate that the main nitrogen-bonding configuration in carboxyl MWCNT@PDA and carboxyl MWCNT@PDAt-900 is pyrrolic-N. The relative atomic fraction of the graphitic-N in carboxyl MWCNT@PDAt-900 is 29.01%, which is higher than that of carboxyl MWCNT@PDA, demonstrating an increase in the concentration of graphitic-N after heat treatment. The graphitic-N shows high catalytic activity and superior stability in $VO^{2+}/VO_2^+$ reaction as reported in the literature [29,37]. Hence, the increase in the relative content of graphitic-N is a benefit for the improvement in electrode performance.

## 3.6. Cyclic voltammetry test

CV tests were performed to investigate the electrochemical performance of the composite electrode. The prepared electrodes with different heat treatment temperature were tested. The influence of carboxyl MWCNT@PDAt on the reaction kinetics of $VO^{2+}/VO_2^+$ redox couple at a scan rate of 5 mV s$^{-1}$ in 0.1 mol l$^{-1}$ VOSO$_4$ + 2.0 mol l$^{-1}$ H$_2$SO$_4$ is shown in figure 6. The carboxyl MWCNT@PDAt-900/GF

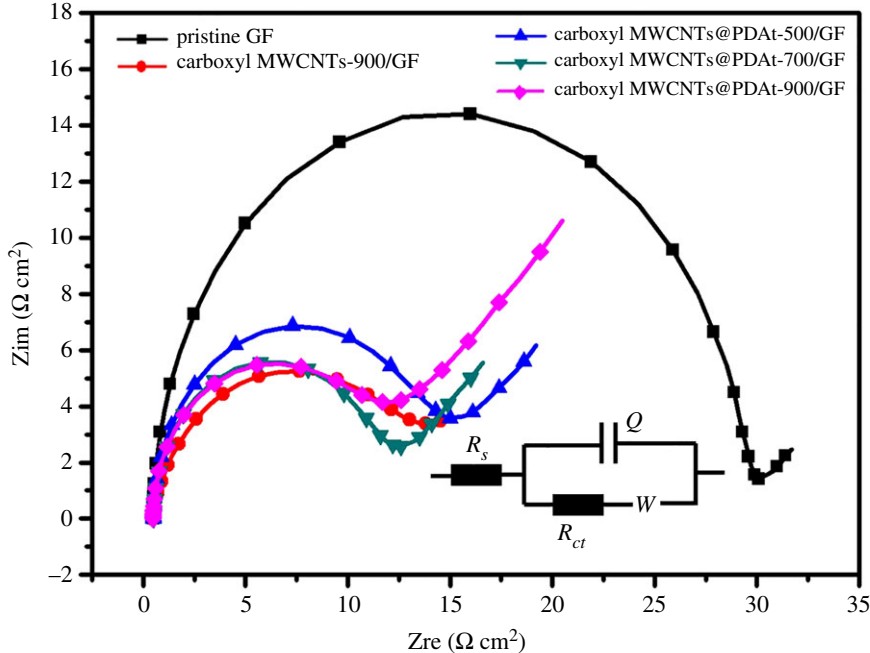

**Figure 7.** EIS tests of pristine GF, carboxyl MWCNT-900/GF and carboxyl MWCNT@PDAt-modified electrodes with different treatment temperatures.

**Table 5.** Electrochemical parameters derived from cyclic voltammetry test curves.

| samples | $E_{pa}$/V | $E_{pc}$/V | $i_{pa}$/A | $i_{pc}$/A | $\Delta E_p$ |
|---|---|---|---|---|---|
| pristine GF | 1.0966 | 0.4996 | 0.1144 | 0.0409 | 0.5970 |
| carboxyl MWCNT-900/GF | 1.0248 | 0.6531 | 0.1143 | 0.0816 | 0.3717 |
| carboxyl MWCNT@PDAt-500/GF | 1.1094 | 0.6509 | 0.1075 | 0.0947 | 0.4585 |
| carboxyl MWCNT@PDAt-700/GF | 1.0518 | 0.6547 | 0.1089 | 0.1016 | 0.3971 |
| carboxyl MWCNT@PDAt-900/GF | 1.0246 | 0.6882 | 0.1421 | 0.1078 | 0.3364 |

exhibits the best kinetics in comparison with other samples. For carboxyl MWCNT@PDAt-900/GF electrode, the anodic and cathodic peaks appear at 1.0246 and 0.6882 eV, respectively. The corresponding anodic peak current ($i_{pa}$) and cathodic peak current ($i_{pc}$) are 0.1421 and 0.1078 A. The increased peak currents indicate the improvement in the reaction kinetics of $VO^{2+}/VO_2^+$ in the presence of carboxyl MWCNT@PDAt on GF surface.

The electrochemical parameters derived from figure 6 are listed in table 5. According to table 5, the peak potential separation ($\Delta E_p$) decreases from 0.5970 V (for pristine GF) to 0.3364 V (for carboxyl MWCNT@PDAt-900/GF), indicating the enhancement in the reversibility of positive redox reaction. The peak potential separations for all carboxyl MWCNT@PDAt-modified electrodes are lower than that for pristine GF, confirming better electrochemical reversibility toward $VO^{2+}/VO_2^+$ reaction. Furthermore, the peak current increases with the heat treatment temperature increasing. The carboxyl MWCNT@PDAt-900/GF shows the highest peak currents compared with other samples. Hence, the decoration of nitrogen-doped carboxyl MWCNT significantly improves the reaction kinetics of $VO^{2+}/VO_2^+$. The reason is owing to the modification of carboxyl MWCNT@PDAt, resulting in high conductivity and large specific surface area, which accelerates the electron transfer rate and provides more active sites for vanadium ions reaction. Additionally, the nitrogen and oxygen containing groups in carboxyl MWCNT@PDAt show high catalytic activity toward $VO^{2+}/VO_2^+$ reaction, effectively facilitating the reaction rate at the electrode/electrolyte interface.

## 3.7. Electrochemical impedance spectroscopy test

EIS tests were also conducted to further evaluate the electrochemical reaction of $VO^{2+}/VO_2^+$ at the surface of carboxyl MWCNT@PDAt-modified electrode. Figure 7 shows the Nyquist plots for the positive redox

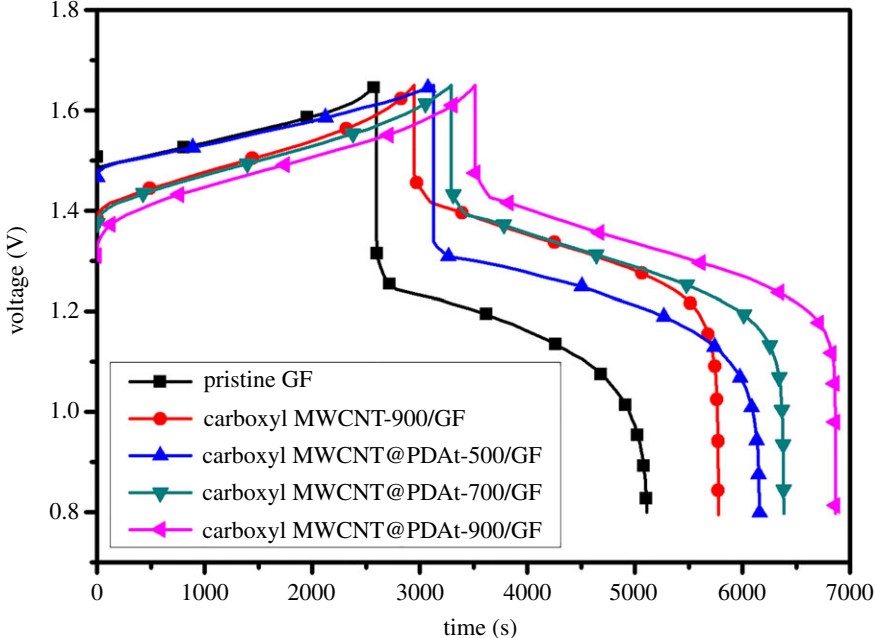

**Figure 8.** Charge/discharge curves of pristine GF, carboxyl MWCNT-900/GF and carboxyl MWCNT@PDAt-modified electrodes with different treatment temperatures.

couple in electrodes with different heat treatment temperature. As seen in figure 7, each Nyquist plot consists of a semicircular part and a linear part, indicating that the $VO^{2+}/VO_2^+$ reaction at all tested electrodes is dominantly controlled by charge transfer and mass transfer process. To further reveal the electrochemical process of $VO^{2+}/VO_2^+$ reaction, the EIS spectra can be fitted by an equivalent circuit shown in figure 7, where $R_s$ represents solution resistance related with electrolyte, $R_{ct}$ reflects the resistance of charge transfer at interface between electrode and electrolyte, $Q$ represents the electric double layer capacitance between electrode and electrolyte interface and the Warburg impedance $W$ is concerned with vanadium ions diffusion. Owing to the poor electrochemical activity of pristine GF, the $R_{ct}$ value of pristine GF is as high as $28.76\,\Omega\,cm^2$. The $R_{ct}$ values of all carboxyl MWCNT@PDAt-modified electrode are much lower compared with that of pristine GF. Hence, the results imply that the electron transfer resistance is reduced after the introduction of carboxyl MWCNT@PDAt. The carboxyl MWCNT@PDAt-900/GF exhibits the lowest value of $R_{ct}$ (approx. $9.98\,\Omega\,cm^2$), indicating a significant enhancement in catalytic activity toward $VO^{2+}/VO_2^+$ reaction.

## 3.8. Charge/discharge test

In order to further confirm the effectiveness of carboxyl MWCNT@PDAt as catalysts in VRFB, a single cell with the prepared electrodes was measured and the charge/discharge results are presented in figure 8. Galvanostatic charge/discharge tests were carried out at a current density of $80\,mA\,cm^{-2}$ with a flow rate of $30\,ml\,min^{-1}$. The corresponding lower and upper voltage for charge/discharge tests were 0.80 and 1.65 V, respectively. In comparison with cell equipped with pristine GF, for the cell with carboxyl MWCNT@PDAt-modified GF charge starts at a lower voltage and discharge starts at a higher voltage. The reason is ascribed to the reduction in electrochemical polarization in the presence of carboxyl MWCNT@PDAt-modified electrodes with improved electrochemical activity and enhanced reaction reversibility.

The detail efficiencies, including current efficiency (CE), voltage efficiency (VE) and energy efficiency (EE) calculated from charge/discharge tests, are listed in table 6. The current efficiencies for all tested samples are higher than 95%, implying that the assembled single cell has a good tightness. The voltage efficiency for cell with pristine GF is 72.05% while that for cell with carboxyl MWCNT@PDAt-900/GF is 84.28%. Owing to the increase in voltage efficiency, the cell with carboxyl MWCNT@PDAt-900/GF shows the highest energy efficiency of 80.54% at a current density of $80\,mA\,cm^{-2}$. The improved cell performance is ascribed to the synergistic effect of nitrogen dopant and oxygen

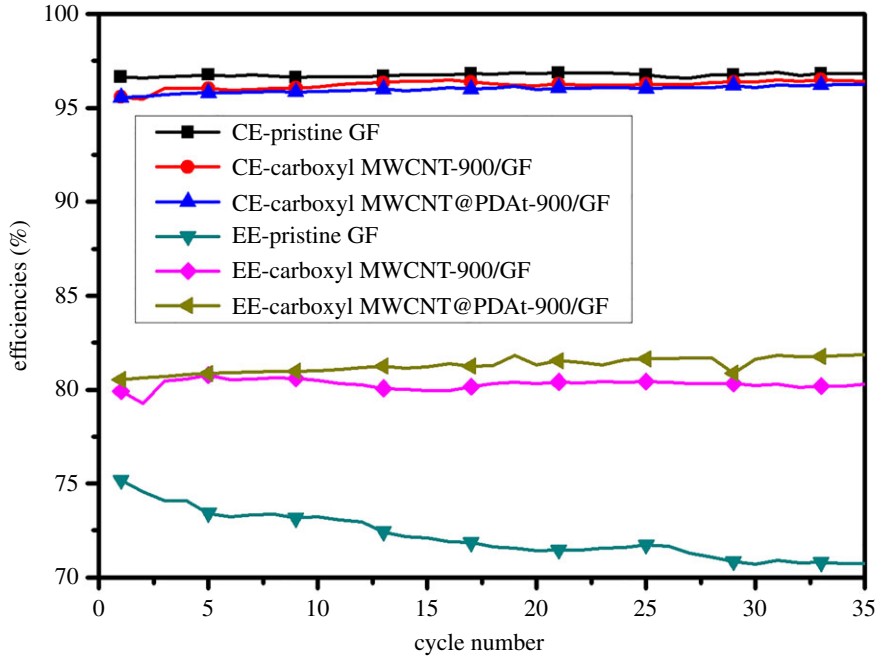

**Figure 9.** Variation of current efficiency and energy efficiency with cycle number for cells assembled with pristine GF, carboxyl MWCNT-900/GF and carboxyl MWCNT@PDAt-900/GFs.

**Table 6.** Efficiencies of cells assembled with different modified electrodes.

| samples | onset potential of charge (V) | onset potential of discharge (V) | current efficiency (%) | voltage efficiency (%) | energy efficiency (%) |
|---|---|---|---|---|---|
| pristine GF | 1.51 | 1.36 | 96.97 | 72.05 | 69.87 |
| carboxyl MWCNT-900/GF | 1.38 | 1.47 | 96.06 | 83.16 | 79.98 |
| carboxyl MWCNT@PDAt-500/GF | 1.47 | 1.35 | 96.96 | 75.76 | 73.46 |
| carboxyl MWCNT@PDAt-700/GF | 1.37 | 1.44 | 95.49 | 80.05 | 76.44 |
| carboxyl MWCNT@PDAt-900/GF | 1.31 | 1.49 | 95.56 | 84.28 | 80.54 |

containing groups introduced by the prepared nitrogen-doped carboxyl MWCNT, resulting in high catalytic activity and enhanced reversibility toward vanadium ions reaction.

## 3.9. Cycling stability test

Figure 9 shows the variations in CE and EE for cells with pristine GF, carboxyl MWCNT-900/GF and carboxyl MWCNT@PDAt-900/GF during 35 cycles. No significant difference in current efficiency is found after 35 cycles for all cells. Compared with cells using carboxyl MWCNT-900/GF, the cell assembled with carboxyl MWCNT@PDAt-900/GF shows a higher energy efficiency and almost no decay in energy efficiency after 35 cycles, demonstrating an excellent VRFB performance. However, the energy efficiency for cell with pristine GF decays from 75.01% to 69.87% after 35 cycles. The improved stability in energy efficiency for carboxyl MWCNT@PDAt-900/GF-based single cell is ascribed to the reduction in electrochemical polarization during charge and discharge processes.

The rate performance of cell with carboxyl MWCNT@PDAt-900/GF at current densities from 80 to 160 mA cm$^{-2}$ is shown in figure 10. It is obvious that the discharge capacity decreases with current density increasing. The reason is ascribed to the higher polarization at a higher current density. The cell with carboxyl MWCNT@PDAt-900/GF exhibits a higher discharge capacity than that with pristine GF at each current density. Especially, the capacity of cell with carboxyl MWCNT@PDAt-900/GF at the current density of 160 mA cm$^{-2}$ is 545.4 mAh, which is 163.4 mAh higher than that with pristine GF. Therefore, VRFB based on carboxyl MWCNT@PDAt-900/GF shows enhanced rate capacity compared

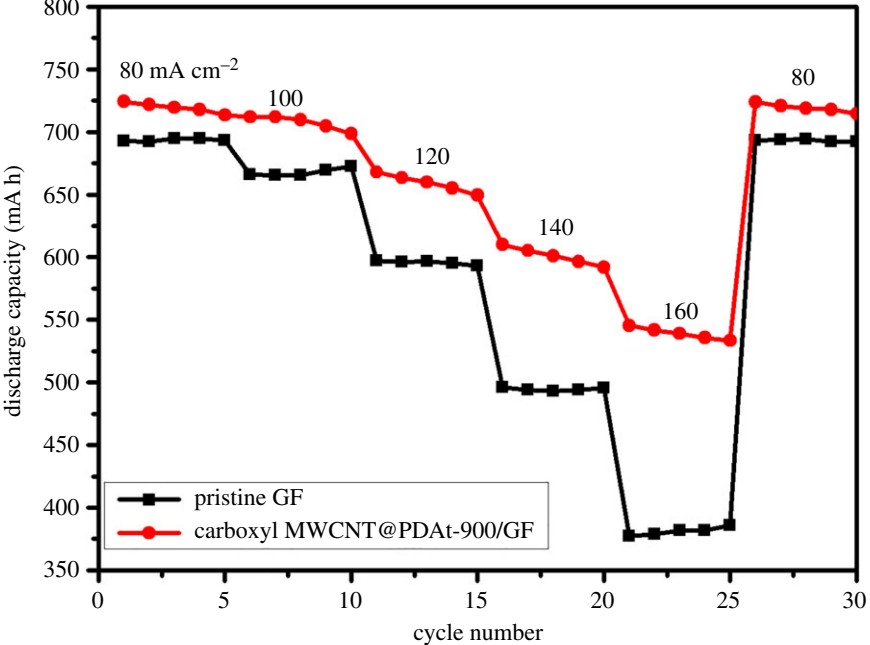

**Figure 10.** Rate performance at current densities from 80 to 160 mA cm$^{-2}$.

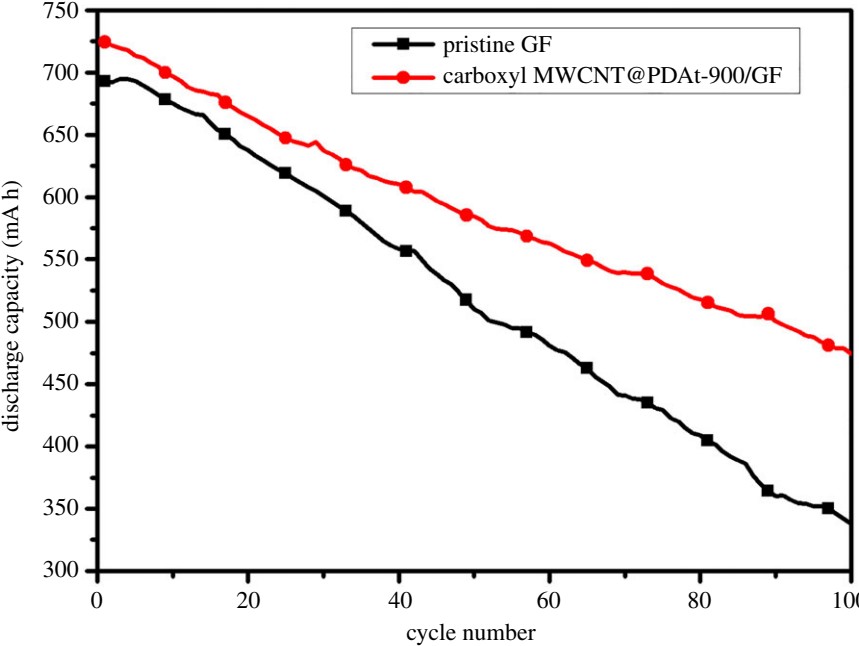

**Figure 11.** Variation of discharge capacity with cycle number under pristine GF and carboxyl MWCNT@PDAt-900/GF.

with that based on pristine GF, which may be owing to the significantly reduced mass transfer and charge transfer resistances resulting from the introduced oxygen and nitrogen containing functional groups.

The variation in discharge capacity during 100 cycles of charge/discharge tests at a current density of 80 mA cm$^{-2}$ is shown in figure 11. With cycling number increasing, a significant decrease in capacity is found for cells with pristine GF and carboxyl MWCNT@PDAt-900/GF. The decrease in capacity is primary owing to the transfer of a small amount of vanadium ions across the proton exchange membrane during long-term redox reaction [38]. The discharge capacity for cell with carboxyl MWCNT@PDAt-900/GF reduces from 724.3 to 474.5 mA h after 100 cycles. As a comparison, the discharge capacity for cell with pristine GF decreases from 692.8 to 337.8 mA h. Therefore, the carboxyl MWCNT@PDAt-900/GF-based cell shows excellent capacity retention after 100 cycles, indicating a relatively high cycling stability in acid solution. Hence, carboxyl MWCNT@PDA-900 as

catalysts contribute to the improvement in discharge capacity and impressive cell performance. However, compared with the cell assembled with pristine GF, the discharge capacity of the cell with carboxyl MWCNT@PDA-900 is only slightly increased, implying that the utilization of electrolyte is still limited. Further efforts should be devoted to the improvement in battery capacity and utilization of electrolyte in the future work.

## 4. Conclusion

Carboxyl MWCNT@PDAt with excellent electrochemical performance were employed as catalysts for VRFB system. Cyclic voltammetry and electrochemical impedance spectroscopy reveal faster reaction kinetics toward $VO^{2+}/VO_2^+$ on the surface of carboxyl MWCNT@PDA-900/GF electrode. The VRFB based on carboxyl MWCNT@PDAt-900 electrode exhibits superior cell performance in terms of high energy efficiency (80.54% at 80 mA cm$^{-2}$) and superior capacity retention. The enhancement of cell performance is attributed to the introduction of nitrogen-doped carboxyl MWCNT using dopamine as an eco-friendly nitrogen source, resulting in a reduction of electrochemical polarization and an increase in reaction kinetics. Therefore, carboxyl MWCNT@PDA-900 as catalysts show promising application in VRFB systems.

Ethics. Research ethics: We were not required to complete an ethical assessment prior to conducting our research. Animal ethics: We were not required to complete an ethical assessment prior to conducting our research.

Permission to carry out fieldwork. No permissions were required prior to conducting our research.

Data accessibility. The datasets supporting this article have been uploaded as part of the electronic supplementary material.

Authors' contributions. Q.L. proposed the conceptualization, methodology and drafted the manuscript. A.B. performed the investigation and analysed the data. T.Z. reviewed the manuscript. S.L. prepared the materials for experiments. H.S. revised the manuscript and funded this project.

Competing interests. There are no conflicts to declare.

Funding. This research is supported by the National Natural Science Foundation of China (51902211 and 51776131), Liaoning Revitalization Talents Program (XLYC1802045) and the Foundation from Liaoning Province of China (20170540746, LJZ2016013 and lnjc201907).

Acknowledgements. We thank the Test Centers of Shenyang Jianzhu University and Xi'an Jiaotong University for materials characterization. We are also grateful to Jie Li, Zhichao Xue and Xiaochen Zhang, who provided meaningful comments that substantially improved the quality of manuscript.

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
