## [Reviewer comments · Royal Society Open Science]

Review History

RSOS-200402.R0 (Original submission)

Review form: Reviewer 1

Is the manuscript scientifically sound in its present form?

Yes

Are the interpretations and conclusions justified by the results?

Yes

Is the language acceptable?

Yes

Do you have any ethical concerns with this paper?

Yes

Have you any concerns about statistical analyses in this paper?

Yes

Recommendation?

Major revision is needed (please make suggestions in comments)

Comments to the Author(s)

The authors reported a dopamine-derived nitrogen doped carboxyl MWCNT modified graphite felt with improved electrochemical activity for VRFB. This reviewer thinks that the proposed strategy can be applied for a rational and effective design of high performance VRFB. Thus, this reviewer strongly recommends the publication of the manuscript in the Royal Society Open Science after a major revision.

1. The authors should provide cycle result more than 100 cycles.
2. The authors should provide rate capability result.

Review form: Reviewer 2

Is the manuscript scientifically sound in its present form?

Yes

Are the interpretations and conclusions justified by the results?

Yes

Is the language acceptable?

Yes

Do you have any ethical concerns with this paper?

No

Have you any concerns about statistical analyses in this paper?

No

Recommendation?

Accept with minor revision (please list in comments)

Comments to the Author(s)

This paper presented the nitrogen and sulfur co-doped reduced graphene oxide as catalyst for VRFB application. Thiourea, which is economical, was used as nitrogen and sulfur sources. The manuscript is well organized and the investigation is innovative. However, there are still several problems needed to be solved.

1. The used heat treatment temperature for the composite electrode is 500, 700 and 900 °C, respectively. Why not treat the NS-rGO/GF at a higher temperature? A higher treatment temperature may lead to a higher degree of carbonization, resulting in an improvement in conductivity.
2. What is the effect of sulfur element on the improved performance of composite electrode?
3. Why the co-doping of nitrogen and sulfur elements has a synergistic effect on the improvement in catalytic activity, resulting in the enhancement in cell performance? Please give more detail explanation.
4. There are several mistakes in the paragraph. For instance, "0.01 to 100 kHz" should be modified to "0.01 to 1×10⁵ Hz". Additionally, "oxidized sulfur groups (-SO_n-)" should be revised to "oxidized sulfur groups (-SO_n-)"
5. Do you think rGO-NS can be well dispersed in DMF at such a high concentration (100 mg rGO-NS in 50 mg DMF) ? Ask the author to give some reasonable feedback.

Decision letter (RSOS-200402.R0)

15-Apr-2020

Dear Dr Li:

Title: Dopamine-derived nitrogen doped carboxyl MWCNT modified graphite felt with improved electrochemical activity for vanadium redox flow batteries
Manuscript ID: RSOS-200402

The editor assigned to your manuscript has now received comments from reviewers. We would like you to revise your paper in accordance with the referee and Subject Editor suggestions which can be found below (not including confidential reports to the Editor). Please note this decision does not guarantee eventual acceptance.

Please submit your revised paper before 08-May-2020. Please note that the revision deadline will expire at 00.00am on this date. If we do not hear from you within this time then it will be assumed that the paper has been withdrawn. In exceptional circumstances, extensions may be possible if agreed with the Editorial Office in advance. We do not allow multiple rounds of revision so we urge you to make every effort to fully address all of the comments at this stage. If deemed necessary by the Editors, your manuscript will be sent back to one or more of the original reviewers for assessment. If the original reviewers are not available we may invite new reviewers.

On behalf of the Subject Editor Professor Anthony Stace and the Associate Editor Dr Darren Walsh.

RSC Associate Editor:
Comments to the Author:
(There are no comments.)

RSC Subject Editor:
Comments to the Author:
(There are no comments.)

Reviewers' Comments to Author:
Reviewer: 1

Comments to the Author(s)

The authors reported a dopamine-derived nitrogen doped carboxyl MWCNT modified graphite felt with improved electrochemical activity for VRFB. This reviewer thinks that the proposed strategy can be applied for a rational and effective design of high performance VRFB. Thus, this reviewer strongly recommends the publication of the manuscript in the Royal Society Open Science after a major revision.

1. The authors should provide cycle result more than 100 cycles.
2. The authors should provide rate capability result.

Reviewer: 2

Comments to the Author(s)

This paper presented the nitrogen and sulfur co-doped reduced graphene oxide as catalyst for VRFB application. Thiourea, which is economical, was used as nitrogen and sulfur sources. The manuscript is well organized and the investigation is innovative. However, there are still several problems needed to be solved.

1. The used heat treatment temperature for the composite electrode is 500, 700 and 900 °C, respectively. Why not treat the NS-rGO/GF at a higher temperature? A higher treatment temperature may lead to a higher degree of carbonization, resulting in an improvement in conductivity.
2. What is the effect of sulfur element on the improved performance of composite electrode?
3. Why the co-doping of nitrogen and sulfur elements has a synergistic effect on the improvement in catalytic activity, resulting in the enhancement in cell performance? Please give more detail explanation.
4. There are several mistakes in the paragraph. For instance, "0.01 to 100 kHz" should be modified to "0.01 to 10⁵ Hz". Additionally, "oxidized sulfur groups (-SO_n-)" should be revised to "oxidized sulfur groups (-SO_n-)"
5. Do you think rGO-NS can be well dispersed in DMF at such a high concentration (100 mg rGO-NS in 50 mg DMF) ? Ask the author to give some reasonable feedback.

Author's Response to Decision Letter for (RSOS-200402.R0)

See Appendix A.

Decision letter (RSOS-200402.R1)

13-May-2020

Dear Professor Sun:

Title: Dopamine-derived nitrogen doped carboxyl MWCNT modified graphite felt with improved electrochemical activity for vanadium redox flow batteries
Manuscript ID: RSOS-200402.R1

It is a pleasure to accept your manuscript in its current form for publication in Royal Society Open Science. The chemistry content of Royal Society Open Science is published in collaboration with the Royal Society of Chemistry.

On behalf of the Subject Editor Professor Anthony Stace and the Associate Editor Dr Darren Walsh.

RSC Associate Editor
Comments to the Author:
(There are no comments.)

Reviewer(s)' Comments to Author:

Appendix A

Response to Referees

Reviewer: 1

Q1. The authors should provide cycle result more than 100 cycles.

Reply:

Thank for your review. Due to the influence of COVID-19, most experiments are restricted. I have done my best to conduct the cycle performance of the cell equipped with the proposed composite electrode. The cycle result more than 100 cycles is performed in the resubmitted manuscript and the details are shown as follows:

The variation in discharge capacity during 100 cycles of charge/discharge tests at a current density of 80 mA/cm² is shown in Fig. 1. With cycling number increasing, a significant decrease in capacity is found for cells with pristine GF and carboxyl MWCNT@PDA-900/GF. The decrease in capacity is primary owing to the transfer of a small amount of vanadium ions across the proton exchange membrane during long-term redox reaction [38]. The discharge capacity for cell with carboxyl MWCNT@PDA-900/GF reduces from 724.3 mAh to 474.5 mAh after 100 cycles. As a comparison, the discharge capacity for cell with pristine GF decreases from 692.8 mAh to 337.8 mAh. Therefore, the carboxyl MWCNT@PDA-900/GF based cell shows excellent capacity retention after 100 cycles, indicating a relatively high cycling stability in acid solution. Hence, carboxyl MWCNT@PDA-900 as catalysts contribute to the improvement in discharge capacity and impressive cell performance. However, compared with the cell assembled with pristine GF, the discharge capacity of the cell with carboxyl MWCNT@PDA-900 is only slightly increased, implying that the utilization of electrolyte is still limited. Further efforts should be devoted to the improvement in battery capacity and utilization of electrolyte in the future work.

Figure 1. Variation of discharge capacity with cycle number under pristine GF and carboxyl MWCNT@PDA-900/GF.

[38] Luo Q, Li L, Wang W, Nie Z, Wei X, Li B, Chen B, Yang Z, Sprenkle V. 2013 Capacity Decay and Remediation of Nafion-based All-Vanadium Redox Flow Batteries. *ChemSusChem* 6, 268-274. (doi: 10.1002/cssc.201200730)

Q2. The authors should provide rate capability result.

Reply:

The rate capability for cell with carboxyl MWCNT@PDA-900 has been tested at current densities from 80 to 160 mA/cm², as shown in Fig. 2. The details are shown as follows:

The rate performance of cell with carboxyl MWCNT@PDA-900/GF at current densities from 80 to 160 mA/cm² is shown in Fig. 2. It is obvious that the discharge capacity decreases with current density increasing. The reason is ascribed to the higher polarization at a higher current density. The cell with carboxyl MWCNT@PDA-900/GF exhibits an increased discharge capacity than that with pristine GF at each current density. Especially, the capacity of cell with carboxyl MWCNT@PDA-900/GF at the current density of 160 mA/cm² is 545.4 mAh, which is 163.4 mAh higher than that with pristine GF. Therefore, VRFB based on carboxyl MWCNT@PDA-900/GF shows enhanced rate capacity than that based on pristine GF, which may be owing to the significantly reduced mass transfer and charge transfer resistances resulting from the introduced oxygen and nitrogen containing functional groups.

Figure. 2 Rate performance at current densities from 80 to 160 mA/cm².

Reviewer: 2

Q1. The used heat treatment temperature for the composite electrode is 500, 700 and 900 °C, respectively. Why not treat the NS-rGO/GF at a higher temperature? A higher treatment temperature may lead to a higher degree of carbonization, resulting in an improvement in conductivity.

Reply:

Thank you for your recommendation. In this paper, the used heat treatment temperature were 500, 700 and 900 °C. Even though a higher treatment temperature may lead to a higher degree of carbonization of composite electrode, it also consumes more energy. Furthermore, the electrode performance is not only related to electrode conductivity, but also depended on the catalytic activity of electrode. The functional groups on electrode surface determine the catalytic activity of electrode. During heat treatment process, the decomposition of functional groups may lead to the decrease in active sites. Hence, the electrode performance is jointly controlled by the inherent conductivity and surface functional groups of electrode. Therefore, comprehensively considering of consumed energy, conductivity and amount of functional groups, the highest temperature used in this manuscript was 900 °C.

Q2. What is the effect of sulfur element on the improved performance of composite electrode?

Reply:

Thank you for your review. For the prepared carboxyl MWCNT@PDA modified graphite felt electrode, the main doped element was nitrogen. According to the reported literature and our characterization results, the doped nitrogen element has positive effect on the performance of composite electrode. In detail, the state of nitrogen element in the composite electrode can be divided into pyridinic-N, pyrrolic-N and graphitic-N. Resulting from the analysis of N1s core-level spectra, the relatively atomic fraction of the graphitic-N in carboxyl MWCNT@PDA-900 is 29.01%, which is higher than that of carboxyl MWCNT@PDA. The improved content of graphitic-N is benefit for the enhancement in electrochemical performance [1, 2]. Compared with carboxyl MWCNT, the carboxyl MWCNT@PDA-900 shows a significant increase in the content of nitrogen. In charge-discharge test, the cell equipped with carboxyl MWCNT@PDA-900/GF performs a higher energy efficiency than that equipped with carboxyl MWCNT -900/GF. Therefore, the improved energy efficiency for cell with carboxyl MWCNT@PDA-900/GF is mainly ascribed to the doped nitrogen element, resulting in an improvement in kinetics of vanadium redox reaction and mass transfer rate of vanadium ions.

[1] J. Jin, X. Fu, Q. Liu, Y. Liu, Z. Wei, K. Niu, J. Zhang, Identifying the Active Site in Nitrogen-Doped Graphene for the $\text{VO}^{2+}/\text{VO}_2^+$ Redox Reaction, *ACS Nano* 7 (2013) 4764-4773.

[2] M. Park, J. Ryu, Y. Kim, J. Cho, Corn protein-derived nitrogen-doped carbon materials with oxygen-rich functional groups: a highly efficient electrocatalyst for all-vanadium redox flow batteries, *Energ. Environ. Sci.* 7 (2014) 3727-3735.

Q3. Why the co-doping of nitrogen and sulfur elements has a synergistic effect on the improvement in catalytic activity, resulting in the enhancement in cell performance? Please give more detail explanation.

Reply:

In this paper, the primary elements in the synthesized composite electrode is nitrogen and oxygen. Owing to the introduction of oxygen functional groups and nitrogen-deficient skeletons, carboxyl MWCNT@PDA-900/GF shows a significant improvement in electrochemical performance. The appearance of N atoms not only

forms vacancies and defects in the composite electrode, but also changes the charge distribution between nitrogen and carbon atoms, resulting in the improvement in mass transfer rate and electron transfer kinetics of vanadium redox couples [1-3]. Furthermore, the oxygen functional group can provide more active sites for vanadium redox reaction, especially the content of C=O is directly related to the number of active sites [4]. During charging process, the positively charged vanadium ions (e.g. VO_2^+) are easily absorbed by the negatively charged nitrogen atoms to form N-V bond. After the oxidation or reduction process, the reaction product (e.g. VO_2^+) is easily diffused from the active site into the electrolyte solution [5]. Therefore, the introduced oxygen functional groups and nitrogen defects in the nitrogen-doped carboxyl MWCNT@PDA-900/GF have a synergistic effect on the enhancement in VRFB performance.

- [1] S. Wang, X. Zhao, T. Cochell, A. Manthiram, Nitrogen-doped carbon nanotube/ graphite felts as advanced electrode materials for vanadium redox flow batteries, *J. Phys. Chem. Lett.* 3 (16) (2012) 2164–2167.
- [2] R. Wang, Y. Li, Twin-cocoon-derived self-standing nitrogen-oxygen-rich monolithic carbon material as the cost-effective electrode for redox flow batteries, *Journal of Power Sources* 421 (2019) 139–146.
- [3] M. Park, I.Y. Jeon, J. Ryu, J.B. Baek, J. Cho, Exploration of the effective location of surface oxygen defects in graphene-based electrocatalysts for all-vanadium redoxflow batteries, *Adv. Energy Mater.* 5 (5) (2015) 1401550.
- [4] M. Park, J. Ryu, Y. Kim, J. Cho, Corn protein-derived nitrogen-doped carbon materials with oxygen-rich functional groups: a highly efficient electrocatalyst for allvanadium redox flow batteries, *Energy Environ. Sci.* 7 (11) (2014) 3727–3735.
- [5] J. Jin, X. Fu, Q. Liu, Y. Liu, Z. Wei, K. Niu, J. Zhang, Identifying the active site in nitrogen-doped graphene for the $\text{VO}_2^+/\text{VO}_2^+$ redox reaction, *ACS Nano* 7 (6) (2013) 4764–4773.

Q4. There are several mistakes in the paragraph. For instance, “0.01 to 100 kHz” should be modified to “0.01 to 1×10^5 Hz”. Additionally, “oxidized sulfur groups (-Son-)” should be revised to “oxidized sulfur groups (-SON-)”.

Reply:

Thank you for your suggestion. The spelling mistakes have been modified in the manuscript and marked in red font. For instance, “0.01 Hz to 100 kHz” has been modified to “0.01 to 1×10^5 Hz”. The sentence “Q represent the electric double layer

capacitance between electrode and electrolyte interface, and the Warburg impedance W concerned with vanadium ions diffusion” has been modified to “ Q represents the electric double layer capacitance between electrode and electrolyte interface, and the Warburg impedance W is concerned with vanadium ions diffusion”.

Q5. Do you think rGO-NS can be well dispersed in DMF at such a high concentration (100 mg rGO-NS in 50 mg DMF)? Ask the author to give some reasonable feedback.

Reply:

Thank you for your review. The prepared catalysts can be well dispersed in DMF at high concentration after ultrasonic treatment for 1h followed by magnetic stirring for 2h, as shown in Fig. 3. No obvious deposition was found after remained at room temperature for 12 h. Therefore, it can be concluded that the catalysts have been uniformly and stably dispersed in DMF.

Figure 3. Catalysts dispersed in DMF.